# Comparison of Depressive Symptoms and Its Influencing Factors among the Elderly in Urban and Rural Areas: Evidence from the China Health and Retirement Longitudinal Study (CHARLS)

**DOI:** 10.3390/ijerph18083886

**Published:** 2021-04-07

**Authors:** Haixia Liu, Xiaojing Fan, Huanyuan Luo, Zhongliang Zhou, Chi Shen, Naibao Hu, Xiangming Zhai

**Affiliations:** 1School of Public Health and Management, Binzhou Medical University, No. 346, Guanhai Road, Yantai 264003, China; liuhaixia127@163.com (H.L.); hnb1981@126.com (N.H.); 2School of Public Health, Xi’an Jiaotong University Health Science Center, No. 76, Yantai West Road, Xi’an 710061, China; 3School of Public Policy and Administration, Xi’an Jiaotong University, No. 28, Xianning West Road, Xi’an 710049, China; fanxj112@xjtu.edu.cn (X.F.); shenchi@stu.xjtu.edu.cn (C.S.); xiangmingzhai0202@163.com (X.Z.); 4Department of Clinical Sciences, Liverpool School of Tropical Medicine, Liverpool L3 5QA, UK; 238551@lstmed.ac.uk

**Keywords:** elderly, depressive symptoms, difference of urban and rural area, CHARLS (wave 4)

## Abstract

Depression amongst the elderly population is a worldwide public health problem, especially in China. Affected by the urban–rural dual structure, depressive symptoms of the elderly in urban and rural areas are significantly different. In order to compare depressive symptoms and its influencing factors among the elderly in urban and rural areas, we used the data from the fourth wave of the China Health and Retirement Longitudinal Study (CHARLS). A total of 7690 participants at age 60 or older were included in this study. The results showed that there was a significant difference in the prevalence estimate of depression between urban and rural elderly (χ^2^ = 10.9.76, *p* < 0.001). The prevalence of depression among rural elderly was significantly higher than that of urban elderly (OR_-unadjusted_ = 1.88, 95% CI: 1.67 to 2.12). After adjusting for gender, age, marital status, education level, minorities, religious belief, self-reported health, duration of sleep, life satisfaction, chronic disease, social activities and having income or not, the prevalence of depression in rural elderly is 1.52 times (OR = 1.52, 95% CI: 1.32 to 1.76) than that of urban elderly. Gender, education level, self-reported health, duration of sleep, chronic diseases were associated with depression in both urban and rural areas. In addition, social activities were connected with depression in urban areas, while minorities, marital status and having income or not were influencing factors of depression among the rural elderly. The interaction analysis showed that the interaction between marital status, social activities and urban and rural sources was statistically significant (divorced: coefficient was 1.567, *p* < 0.05; social activities: coefficient was 0.340, *p* < 0.05), while gender, education level, minorities, self-reported health, duration of sleep, life satisfaction, chronic disease, social activities having income or not and urban and rural sources have no interaction (*p* > 0.05). Thus, it is necessary to propose targeted and precise intervention strategies to prevent depression after accurately identifying the factors’ effects.

## 1. Introduction

According to the data of China’s National Bureau of Statistics, by the end of 2019, there were 253.88 million elderly people aged 60 years and over in China, accounting for 18.1% of the total population. Predication indicated that by 2050, the elderly population in China will increase substantially, with up to 400.83 million people aged 60 years and over [1,2]. With the aging problems becoming increasingly serious, more attention must be paid to the health of the elderly. Depression amongst the elderly is a global public health problem [3]. The Global Health Estimates reported by the World Health Organization (WHO) estimated that the prevalence of depression had peaked amongst the elderly [4]. More than 10% of the elderly worldwide have experienced depressive symptoms. While in China, the prevalence rate of depression in the elderly is 23.6%, which is higher than that of other age groups [5].

Depression is a common psychological disorder, usually manifested as sadness, self-depression, loss of interest in things and other negative emotions [6,7,8]. Studies have shown that when people transition from middle adulthood to older age, depressive symptoms tend to increase, often together with worsening physical health [9,10,11], which may be a main cause of disabilities and a big contributor to the global disease burden. In China, depression is the fourth leading cause of disabilities and affects 54 million people as estimated by WHO [12,13]. Meanwhile, the development of depression had group heterogeneity, and locus forms of depressive symptoms vary by people’s characteristics [14,15,16,17]. The identification of heterogeneous locus is helpful to further analyze the causes of depression in the elderly, and the heterogeneous locus model can provide ideas for related interventions for elderly depressive symptoms [17].

Depression is a prevalent psychiatric disorder associated with biological, social factors, sensory processing patterns and functional impairment [18,19,20,21]. Gianluce S et al.’s study found that extreme sensory processing patterns was an important factor contributing to the complex pathophysiology of major depression. Lower registration of sensory input referring to hypo-sensitivity and sensation avoiding referring to hypersensitivity significantly correlated with higher alexithymia and, in particular, with difficulties to describe and identify feeling. Lower ability to register sensory input was an important factor involved in determining depression [21]. Some studies showed that gender, age, marital, education level, chronic disease, sleep, life satisfaction, social activities, health and quality of life were influencing factors of depression in the elderly. For example, Zhang L’s study found gender was an influencing factor of depression among elderly, the average level of depressive symptom for female elderly was higher than that of male elderly [22]. Li JS’s study found that the elderly with lower education level, less social activities, more chronic diseases were more likely to have depression symptoms [23]. Feng’s study found chronic disease was an important influencing factor of depressive symptoms in the elderly, the incidence of depressive symptoms in the elderly with more than two or three chronic diseases is high [24,25]. Wu S’s study found that age and duration of sleep were the main factors affecting the life satisfaction and depression of the elderly in China. The lower the life satisfaction, the higher the prevalence estimates of depression. Further, the age and duration of sleep were important protective factor of depression in the elderly [26]. Li Y’s study indicated that social activities was an important influencing factor of the depressive symptoms of the elderly, the prevalence of elderly having social activities was higher than that of without social activities, and the association between social participation and depressive symptoms at old age varies by gender and by urban and rural areas [27,28,29]. Marital status was an important influencing factor of depression. The effects of divorce and never married on depression of the elderly were different. Divorce, in particular had a greater impact on depression in the elderly [30,31,32]. Xu et al. and Harithasan et al.’s studies indicated that health and quality of life were associated with the depressive symptoms of elderly [7,18,19].

At present, many theories have been introduced to explain the mechanism of urbanization’s influence on mental health, such as Grossman’s theory of healthy production, Michael Marmot’s social determinism, Max Weber’s theory of social stratification, the social decision theory of health [33,34] and Blum’s model of environmental health medicine. Blum’s model of environmental health medicine proposed that environment factors, especially the social environment, played an important role in people’s health, physical and spiritual development [34]. Urbanization is an important manifestation of the current social and economic environment, which affects residents’ health by changing the social environment. With the acceleration of China’s urbanization, the imbalance in social and economic development between urban and rural areas affects the difference in depressive symptoms among the elderly in urban and rural areas. According to the 1% sample survey data of China’s population in 2005, 25.5% of the rural elderly suffered from moderate and severe depression, which was much higher than that of the elderly in urban areas (13.6%) [34]. Based on the data of 2013 China Health and Retirement Longitudinal Study (CHARLS, provided by the National Development Research Institute of Peking University), Yang et al.’s study in 2013 found that the prevalence rates of “elderly depression” in urban and rural areas were 16.3% and 30.0% respectively [35]. Using data of CHARLS (2015 y), Shan et al. found that the depression prevalence was nearly 30.08% among middle-aged and elderly people, and the higher the level of urbanization, the lower the prevalence of depression [13,30,36,37]. Gan et al.’s study (1075 elderly people aged 65 years and over) indicated that the prevalence rate of the urban elderly (23.5%) was lower than that of the rural elderly (31.9%) [38]. He and Zhao et al. studied the association of Chinese drifting elderly intergeneration support satisfaction with expectation in Shanghai, China, and found that the urban and rural elderly have received different intergenerational support which had a very important impact on depression, especially for the drifting elderly [39,40,41].

In conclusion, urbanization is an important influencing factor of elderly depression. The imbalance between urban and rural areas leads to heterogeneous loci of depression in urban and rural elderly. However, the association of urbanization with depression is not clear enough, and the specific differences in depressive symptoms and its influencing factors between urban and rural elderly are also unclear. Moreover, the elderly age structure, gender structure, education level, social activities and other situations are different between urban and rural areas [22,27]. In order to control these variables, binary logistics regression was employed to estimate the association between urban or rural sources and depression, also compare the influencing factors of depression in urban and rural elderly respectively, and put forward the following two hypotheses.

**Hypothesis** **1.**
*Urban and rural source is an important influencing factor of depression in the elderly. There are significant differences in the prevalence estimates of depression between urban and rural elderly, and the prevalence of depression in the elderly in rural areas is higher than that in urban areas.*


**Hypothesis** **2.**
*The influencing factors of depressive symptoms in urban and rural elderly are different. Even if the influencing factors are the same, the impacts of these factors on depressive symptoms are different in urban and rural areas.*


## 2. Material and Methods

### 2.1. Participants and Design

Data were obtained from the fourth wave survey data from the China Health and Retirement Longitudinal Study (CHARLS, wave 4), which was the latest data of CHARLS released in September 2020. CHARLS is a survey of middle-aged and elderly people in China, based on a sample of households with members aged 45 years or above. It aims to establish a high quality public microdatabase that can provide a wide range of information from socioeconomic status to health conditions, to meet the needs of scientific research on the middle-aged and elderly people [38,41]. To ensure sample representativeness, the survey followed strict randomization procedures and used a multi-stage sampling method. When sampling county and rural administrative units, a probability proportional to size (PPS) sampling mothed was adopted [40,41,42,43]. In the first stage of sampling, 150 county-level units were randomly selected using the PPS method from a sampling frame containing all county-level units in China (excluding Tibet). In the second stage, three communities (rural administrative villages or urban resident committees) were randomly chose using the PPS method from a sampling frame containing all communities in the county-level units. In the third stage, to create a sampling frame, using the software developed by the CHARLS team which utilized Google Earth map images, all dwelling units in a community were listed following an extensive mapping and listing operation, and then a certain number of dwelling units were randomly chose [39,40,41,42,43]. All data collected in CHARLS are maintained by the Institute of Social Science Survey of Peking University and have been publicly released on the CHARLS website (http://charls.pku.edu.cn/pages/data/2018-charls-wave4/zh-cn.html, accessed on 28 December 2020).

In order to understand the recent situation of depressive symptoms in the elderly, we used data of Wave 4 (2018 y) of the CHARLS, which released in September 2020, involved 19,816 respondents in 150 counties/districts and 450 villages/urban communities. According to the study purpose, the elderly aged 60 years and above were selected as the research objects. According to the research purpose, the elderly aged 60 and above were selected as the research objects. The real age of the elderly was calculated according to the questions in the questionnaire “what is your actual date of birth?” and the survey time (July or August 2018) (the real age = the respondent’s birth year and month minus the interview year and month). A total of 10,107 respondents aged 60 and above were included, of which 8294 completed the depression scale and 1813 did not. According to the variables involved in our study, as long as any one of the variables is missing, the elderly will be excluded. There are 604 elderly people who did not fill in the “urban and rural source” and “sleep time” question. Finally, a total of 7690 elderly people will be included in the analysis (the sample screening analysis framework is shown in Figure 1).

### 2.2. Measurement

Depressive symptoms were measured using the Center for Epidemiologic Studies Depression Scale (CES-D-10) which has been validated among elderly respondents in China using CHARLS data [44]. The CES-D-10 includes 10 questions regarding the participant’s experience in the past week: feeling bothered, having trouble in concentrating, feeling depressed, and so on [45]. The total score ranges from 0 to 30, higher score indicating more severe depressive symptoms. In this study, we used a cutoff score of ≥10 to distinguish participants with depression from those who were relatively free of depression [46,47]. In the first three rounds of CHARLS, the CES-D-10 all had good internal consistency [46], and α = 0.818 in our study, with good reliability too. In our study, depression was the dependent variable, and the demographic variables included urban and rural areas, gender, age, marital status, education level, minorities, religious belief, having income or not. Some health-related variables were also included, such as self-report health, duration of sleep, life satisfaction, chronic disease, and social activities. The descriptions of the variables used in this study are all shown in Table 1.

### 2.3. Statistical Analysis

First, demographic characteristics of the elderly in urban and rural areas were described as frequencies and percentages. The Chi-square test was used to compare the difference in the prevalence of depression between urban and rural elderly. Then, univariate and multivariate logistics regression was used to calculate the unadjusted and adjusted odds ratios (ORs) of the covariates to identify the influencing factors of depressive symptoms among the urban and rural elderly, also the interaction between all covariates and urban and rural sources were calculated to check whether these factors had different effects on depression in urban and rural areas (significance in difference). All statistical analyses were performed using IBM SPSS StatisticsV22.0, and *p*-value < 0.05 was considered statistically significant. 

## 3. Results

### 3.1. Characteristics of the Participants

Among the 7690 participants, urban elderly accounted for about 26%, and rural elderly accounted for 74%. In urban areas, there were slightly less male than female, while the situation in rural areas was the opposite. In both areas, around half of the elderly were 65 to 74 years old and around one third were 60 to 64 years old. The share of oldest age group was somewhat larger in urban areas. In both areas, married and living with spouse accounted for about 75%. Around half of the of the elderly’s self-reported health were fair in both areas, and around one fifth of elderly’s self-reported health were poor and very poor in urban areas, in comparison to one almost one third in rural areas. In urban areas, around 7% of the elderly were not very satisfied and not at all satisfied, while in rural areas this number was 10%. Slightly more than 50% of urban elderly had chronic disease, and the share of rural elderly with chronic disease was around 45%. About 64% of urban elderly had social activities, and about 45% of rural elderly had social activities. In urban areas, around 22% of the elderly had income except pension, while in rural areas this number was 21%. Other characteristics of the participants are all shown in Table 2.

### 3.2. Comparison of Depression in the Elderly between Urban and Rural Areas

The result of Chi-square test showed that there was a significant difference in the prevalence estimate of depression between urban and rural elderly (χ^2^ = 10.9.76, *p* < 0.001), and the prevalence of depression in the rural elderly was higher than that in the urban elderly, which checked Hypothesis 1. The prevalence estimate of depression in rural elderly was 1.88 times than that of urban elderly (OR-unadjusted = 1.88, 95% CI: 1.67 to 2.12). The result of logistics regression showed that urban or rural source was an important influencing factor of depression in the elderly, and the risk of depression in rural elderly was 1.52 times that of urban elderly after adjusting for gender, age, marital status, education level, minorities, religious belief, self-reported health, duration of sleep, life satisfaction, chronic disease, social activities and having income or not (Table 3). These results checked Hypothesis 1. Urban and rural source, gender, marital status (widowed), education level, minorities, duration of sleep, life satisfaction, chronic diseases, social activities and having income or not (except pension) were the influencing factors of depressive symptoms in the elderly. Age, marital status (except for widowed), religious beliefs, self-reported health had no effect on depressive symptoms, which answered Hyphosis 2 (the influencing factors of depressive symptoms in urban and rural elderly are different). Female elderly had higher likelihood of depressive symptoms (OR = 1.62, 95% CI: 1.43 to 1.84). The prevalence estimate of depression among ethnic minorities elderly was higher than Han minority elderly (OR = 1.31, 95% CI: 1.06 to 1.62). The prevalence of depression among widowed elderly was 1.22 times than that of elderly who married and lived with spouse (OR = 1.22, 95% CI: 1.05 to 1.42). Compared with the elderly with middle school or high/vocational school and above education level, the elderly with illiterate education level were more likely to have depression (middle school: *OR* = 0.68, 95%*CI*: 0.56 to 0.83; high/vocational school and above: OR = 0.47, 95% CI: 0.37 to 0.61). The elderly who sleep less than or equal to 5 h were more prone to have depression than those who sleep 6–9 h (OR = 0.48, 95% CI: 0.43 to 0.54) or 10 h or more (OR= 0.53, 95% CI: 0.44 to 0.65).The prevalence of depression in the elderly with chronic diseases was higher than that in the elderly without chronic diseases (OR = 1.22, 95% CI:1.09 to 1.37), and the prevalence of depression in the elderly who had social activities than that in the elderly without social activities (OR = 0.87, 95% CI: 0.77 to 0.97), and the prevalence of the elderly having social activities was lower than that of elderly who had no social activities (OR = 0.87, 95% CI: 0.70 to 0.92). The lower the life satisfaction of the elderly, the more prone to have depression (except for very satisfied/somewhat satisfied: OR = 1.98, 95% CI: 1.48 to 2.64; not very satisfied: OR = 7.39, 95% CI: 5.25 to 10.39; not satisfied at all: OR = 17.11, 95% CI: 10.11 to 28.97).

### 3.3. Comparison of Influencing Factors of Depression between Urban and Rural Areas

Table 4 showed that in urban areas, different gender, marital status, education level, self-reported health, duration of sleep, life satisfaction, chronic disease, social activities group had different prevalence estimates of depression (*p* < 0.05). While in rural areas, the prevalence estimates were different in gender, marital status, education level, minorities, self-reported health, duration of sleep, chronic disease, social activities and income groups (*p* < 0.05). The results of binary logistics regression showed that gender, education level, self-reported health, duration of sleep, chronic diseases were associated with depression in both urban and rural areas (Table 5). However, there were also some different influencing factors. Social activities were connected with depression in urban areas, while minorities, marital status and having income or not were influencing factors of depression among the rural elderly. The prevalence of the urban elderly who had social activities was 0.68 times than that of the elderly without social activities (OR = 0.68, 95% CI: 0.51 to 0.90). The prevalence of ethnic minorities elderly was 1.37 times (OR = 1.37, 95% CI: 1.08 to 1.74) than that of Han nationality elderly in rural areas, the prevalence of the rural elderly having income was 0.79 times than that of without income (OR = 0.79, 95% CI:0.69 to 0.93), and the prevalence of divorced and widowed elderly were higher than that of married and live with spouse elderly (divorced: OR = 2.15, 95% CI:1.01 to 4.58; widowed: OR = 1.24, 95% CI: 1.05 to 1.4). Seeing from the effects of these influencing factors, in urban areas, the prevalence of depression in female elderly was 1.46 times (OR = 1.46, 95% CI: 1.09 to 1.96) than that of male elderly, while in rural areas, the prevalence of depression in female elderly was 1.68 times (OR = 1.68, 95% CI: 1.46 to 1.93) than that of male elderly. Seeing from the impacts of education level, the prevalence of depression among middle school education level elderly in urban areas was 0.55 times (OR = 0.55, 95% CI: 0.34 to 0.88) than that of elderly with illiterate education level, and the prevalence of elderly with high\vocational school and above education level was 0.40 times (OR = 0.40, 95% CI: 0.24 to 0.65) than that of elderly with illiterate education level. While in rural areas, the prevalence of depression among middle school education level elderly was 0.71 times (OR = 0.71, 95% CI: 0.56 to 0.89) than that of elderly with illiterate education level, and the prevalence of elderly with high\vocational school and above education level was 0.42 times (OR = 0.42, 95% CI: 0.29 to 0.61) than that of elderly with illiterate education level. The prevalence of urban elderly who had chronic diseases was 1.68 times than that of elderly without chronic diseases (OR = 1.68, 95% CI: 1.27 to 2.23), while it was 1.17 times in rural areas (OR = 1.17, 95% CI: 1.03 to 1.33). Above all, the influencing factors of depression in urban and rural areas were different. Even if the influencing factors were the same, the impacts of these factors on depression symptoms may be different. And these results answered Hypothesis 2. In order to check Hyphothesis 2, we also have done the interaction analysis between these control variables and urban and rural sources using logistics regression. The results of interaction analyses (significance of difference in Table 5) showed only the interaction between marital status, social activities and urban and rural sources was statistically significant (divorced: coefficient was 1.567, *p* < 0.05; social activities: coefficient was 0.340, *p* < 0.05), which indicated that there was interaction between marital status (divorced), social activities and urban and rural, and the impacts on elderly depression was different between urban and rural area. The interaction between gender, age, education level, minorities, religious beliefs, self-reported health, duration of sleep, life satisfaction, chronic disease, having income or not and urban and rural sources were not significant, the impacts of these variables on urban and rural elderly depression were not significant different (*p* > 0.05).

## 4. Discussion

Our research found that there was significant difference in the prevalence rate of depression between urban and rural elderly. The depression estimate in rural elderly was significantly higher than that of the urban elderly. The result of logistics regression showed that the risk of depression in rural elderly was 1.52 times that of urban elderly after adjusting for gender, age, marital status, education level, minorities, religious belief, self-reported health, duration of sleep, life satisfaction, chronic disease, social activities and having income or not, which answered Hypothesis 1 (urban and rural sources was an important influencing factor of depression in the elderly. There were significant differences in the prevalence estimates of depression between urban and rural elderly, and the prevalence of depression in the elderly in rural areas was higher than that in urban areas). Urbanization, an important influencing factor of the current socio-economic environment, which leads to the imbalance of social and economic development between urban and rural areas and the difference in depressive symptoms in urban and rural elderly. Studies, mainly conducted in China, found urbanization can have protective effect on mental health, considering that cities are also associated with improved infrastructures, more resources, more opportunity and better social support and welfare [45,46,47]. While other studies, including in the United States, the India, the Netherlands and Vietnam found that urbanization was detrimental to mental health, because cities are overcrowded, have less greens and space, involve pollution and traffic noise, and so on [48,49,50]. Therefore, we should pay more attention to the mental health problems of elderly, accurately identify the characteristics, and provide targeted psychological intervention and health care measures according to the influencing factors and their impacts. In both urban and rural areas, the prevalence of depression varies significantly among the elderly by characteristics, such as gender, marital status, chronic disease, education level, duration of sleep, self-reported health and life satisfaction [51,52,53]. The prevalence of depression in females was higher than that in males, which was consistent with Yang et al.’s study using CHARLS data in 2013 [35]. Although females survive for a long time, compared with males, they are in a weak position in terms of health level. The main reasons are that the female elderly live longer, have a higher rate of widowhood, and have a lack of companionship, thus, inner loneliness aggravates it. In addition, economic independence in females is poor, especially in rural areas, many of them are dependent on male, with less social contact, and the difference of intergenerational support is also easy to cause the difference of depression between males and females [25,26,27]. Zhang L’s study found that the male elderly got more financial support than elderly women, and female elderly take care of their grandchildren longer than the male elderly, which lead to more serious depression of the female elderly [21,27,28]. Therefore, we should focus on the mental health problems of the female elderly, give them more help and care in the prevention of chronic diseases, family division of labor, social support and other aspects, and encourage them to go out to participate in more social activities and communication. From the perspective of longitudinal comparison, the prevalence of depression of the elderly in urban and rural areas in our study (CHARLS in 2018 y) was higher than that of Zhou et al.’s study [31,35] using the CHARLS data in 2013, indicating that the depression prevalence of the elderly in urban and rural China was on the rise in the past five years. Both in urban and rural areas, education level was an important influencing factor of depression among the elderly, which was consistent with Yang’s study [35]. In our study, we found that with the improvement of education level, the prevalence of depression in the elderly gradually decreases (except for the elementary school and below education level), and the effects on depression between urban and rural elderly was not significant. From the perspective of life satisfaction and self-reported health, with the improvement of life satisfaction and self-reported health, the prevalence rates of depression among the elderly decreased in both urban and rural areas. Self-reported health is based on subjective feelings from the elderly about their own health status and it largely depends on the psychological status of the elderly. The better their self-reported health, the more confident they are in their health and the more likely they are to take a positive attitude towards life [24,35]. In our study, we found that after controlling other independent variables, in urban areas, the prevalence of depression of elderly with fair health was 2.50 time than that of elderly with very good health, and the elderly with poor health was 5.98 times than that of very good health elderly, very poor health was 12.74 times than that of very good health. While the impacts of self-reported health on the depression had no significant difference between urban and rural areas (Table 5). Elderly with good health often have good psychological and physiological conditions, and can actively participate in some daily activities, such as square dancing, parent–child communication, University of the elderly, community activities that gather elderly people together, which can improve the social existence and happiness, and maintain a good mentality. Therefore, the relevant departments and elderly families should give more care to the elderly, supervise them to go to the hospital for physical examination regularly, carry out health education for residents, carry out health promotion intervention activities, improve the quality of life and health level of the elderly, reduce their bad emotions, and then improve their life satisfaction level and mental health level [27,28,54,55,56,57,58,59].

Chronic disease is an important factor influencing depressive symptoms of the elderly. In urban areas, the prevalence of elderly people with depressive symptoms was 1.68 times that of elderly people without chronic diseases. While it was 1.17 times in rural areas, there was no significant difference between urban and rural areas. Zhang et al.’s study also found that the prevalence of depression in the elderly with chronic diseases was 2.35 times that of the elderly without chronic diseases, and among 2370 elderly over 60 years old with depression, 56.3% had somatic pain [60]. We also found that having chronic diseases was a risk factor for depression in the elderly. Duration of sleep is also a factor influencing depression of the urban (6–9 h) and rural (6–9 h and ≥10 h) elderly. Not only the duration of sleep had impact on the elderly depression, but also the quality of sleep. Zhang et al.’s study found that the poor quality of sleep can be a component of depression itself. Probably it is not the sleep quality that influences the depressive symptoms, but it is part of the syndrome. The prevalence depression among elderly adults with poor sleep quality (31.1%) was significantly higher than that of elderly adults with good sleep quality (12.3%) [60]. Depressive symptoms in the elderly may be affected by social activities. This study found that both in urban and rural areas, the elderly who engaged in social activities were less likely to suffer from depression, which indicated that social activities was an important influencing factor of depression among elderly. According to the interaction analysis, we found that the impacts on urban and rural areas was significant. Ae et al.’s study found that depression was a risk factor in health-promoting behaviors, and that the engagement level in health-promoting behaviors increased as the depression level decreased in the low-income ordinary elderly hypertensive patients [61,62]. Elderly people with poor functional status are at higher risk of depression in urban areas. All these suggested that improving the social activities of the urban elderly is an important measure to reduce the prevalence rate of depression.

## 5. Limitations

Our study also had some limitations. Firstly, although this study included demographic variables, life satisfaction, social activities and income (except pension), no relevant regional economic level and comprehensive income level factors were analyzed. These factors may also have impacts on the elderly depressive symptoms due to the effects of urbanization. Secondly, although the CHARLS data was representative, the cross-sectional data cannot determine the causal relationship between living in urban or rural areas and depressive symptoms. Therefore, we need to conduct studies on multiple rounds of CHARLS data or experimental studies.

## 6. Conclusions

Despite the limitations, the study results provided some data and basis for clarifying the urban–rural differences and its influencing factors among the elderly between urban and rural areas. We used the nationwide tracking data (CHARLS), the sample size was large and the sample had good representativeness. The results have shown that the prevalence of depression symptoms in rural areas was higher than in urban areas. Gender, education level, self-reported health, duration of sleep, chronic diseases were associated with depression in both urban and rural areas. In addition, social activities were connected with depression in urban areas, while minorities, marital status and having income or not were influencing factors of depression among the rural elderly. Although some of these factors had impact on urban and rural areas, there was no significant difference except for the variables of marital status (divorced) and social activities. Thus, it is necessary to propose targeted and precise intervention strategies to prevent depression after accurately identifying their impact.

## Figures and Tables

**Figure 1 ijerph-18-03886-f001:**
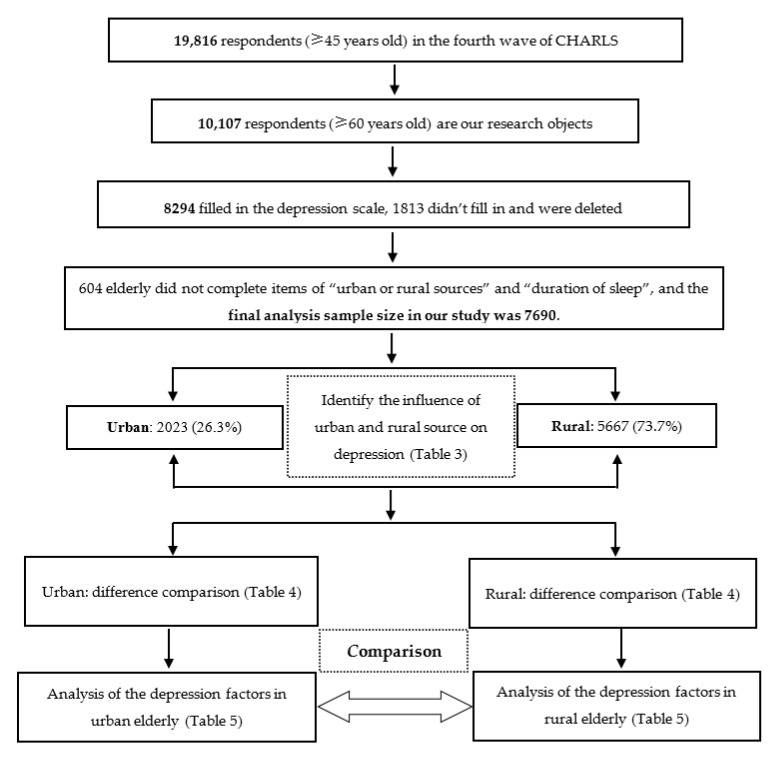
The flowchart of this study.

**Table 1 ijerph-18-03886-t001:** Description of variables used in this study.

Type of Variables	Name and Definition of Variables	Variable Assignment
Demographic variables	Urban and rural areas (their Hukou status, which is a population registration system used in China that indicate individual’s rural or urban residency status. Specifically, agricultural hukou was classified as rural hukou, while non-agricultural hukou and unified residence hukou were classified as urban hukou. Hukou is a population registration system that has long been used in China. Every Chinese citizen is required to legally register in the system, as either agricultural or non-agricultural residency (normally referred to as rural vs. urban)	Urban = 1
		Rural = 2
	Gender	Male = 1
		Female = 2
	Age (the real age = the respondent’s birth year and month minus the interview year and month.)	60–64 = 1
		65–74 = 2
		≥75 = 3
	Marital status	Married and live with spouse = 1
		Married but not live with spouse = 2
		Divorced = 3
		Widower = 4
		Never married = 5
	Education level (the highest level of education that the respondent attained was self-reported in CHARLS)	Illiterate = 1
		Elementary school and below = 2
		Middle school = 3
		High\vocational school and above = 4
	Minorities (there are 56 ethnic groups in China, all of which are called ethnic minorities except the Han nationality)	Han = 1
		Ethnic minorities = 2
	Religious belief	No = 0
		Yes = 1
	Having income or not (did you receive any wage and bonus income [except pension] in the past year?)	No = 0
		Yes = 1
Health related variables	Self-reported health (would you say your health is very good, good, fair, poor or very poor?)	Very good = 1
		Good = 2
		Fair = 3
		Poor = 4
		Very poor = 5
	Duration of sleep (during the past month, how many hours of actual sleep did you get at night? (average hours for one night))	≤5 h = 1
		6–9 h = 2
		≥10 h = 3
	Life satisfaction (How satisfied are you with your life overall? Are you completely satisfied, very satisfied, somewhat satisfied, not very satisfied, or not satisfied at all?)	Completely satisfied = 1
		Very satisfied = 2
		Somewhat satisfied = 3
		Not very satisfied = 4
		Not satisfied at all =5
	Chronic disease (including hypertension, diabetes or high blood sugar, cancer or a malignant tumor, chronic lung disease such as chronic bronchitis or emphysema, heart diseases, stroke, emotional, nervous, or psychiatric problems, arthritis, dyslipidemia, liver disease, kidney disease, stomach or other digestive disease, asthma.)	Having none of these chronic disease = 0
		Having one of these chronic disease = 1
	Social activities (whether they participated in the following social activities in the past month: “interacted with friend”, “played Ma-jong, chess, cards, or went to a community club”, “sent to a sporting event, participated in a social group, or participated in some other sort of club”, “took part in a community-related organization”, “took part in voluntary or charity work”, “attended an educational or training course”.)	Having none of these social activities = 0
		Having one of these social activities = 1
Dependent variable	Depressive symptoms (recoded from continuous variable [1–30 scores], and transformed into binary variable, ≤10: negative; >10: positive)	≤10 = 0
		>10 = 1

**Table 2 ijerph-18-03886-t002:** Characteristics of participants (N (%)).

Variables	Urban	Rural	Total
Gender			
Male	987 (48.8)	2962 (52.3)	3949 (51.4)
Female	1036 (52.30)	2705 (47.7)	3741 (48.6)
Age			
60–64	712 (35.2)	2091 (36.9)	2803 (36.4)
65–74	958 (47.4)	2803 (49.5)	3761 (48.9)
≥75	353 (17.4)	773 (13.6)	1126 (14.6)
Marital status			
Married and live with spouse	1574 (77.8)	4405 (77.7)	5979 (77.8)
Married but not live with spouse	68 (3.4)	201 (3.5)	269 (3.5)
Divorced	35 (1.7)	38 (0.7)	73 (0.9)
Widowed	346 (17.1)	1023 (18.1)	1323 (17.2)
Never married	6 (0.3)	40 (0.7)	46 (0.6)
Education level			
Illiterate	225 (11.1)	1761 (31.1)	1986 (25.8)
Elementary school and below	760 (37.6)	2858 (50.4)	3618 (47.0)
Middle school	547 (27.0)	758 (13.4)	1305 (17.0)
High\vocational school and above	491 (24.3)	290 (5.1)	781 (10.1)
Minorities			
Han	1885 (93.2)	5288 (93.3)	7173 (92.8)
Ethnic minorities	138 (6.8)	397 (6.7)	517 (7.2)
Religious beliefs			
No	1815 (89.7)	5051 (89.1)	6866 (89.3)
Yes	208 (10.3)	616 (10.9)	824 (10.7)
Self-reported health			
Very good	212 (10.5)	592 (10.4)	804 (10.5)
Good	278 (13.7)	586 (10.3)	864 (11.2)
Fair	1072 (53.0)	2710 (47.8)	3782 (49.2)
Poor	351 (17.4)	1368 (24.1)	1719 (22.4)
Very poor	110 (5.4)	408 (7.2)	518 (6.7)
Duration of sleep			
≤5 h	676 (33.4)	2059 (36.3)	2735 (35.6)
6–9 h	1254 (62.0)	2955 (52.1)	4209 (54.7)
≥10 h	93 (4.6)	653 (11.5)	746 (9.7)
Life satisfaction			
Completely satisfied	107 (5.3)	277 (4.9)	384 (5.0)
Very satisfied	605 (29.9)	1873 (33.1)	2478 (32.2)
Somewhat satisfied	1174 (58.0)	2915 (51.3)	4089 (53.2)
Not very satisfied	99 (4.9)	446 (7.9)	545 (7.1)
Not at all satisfied	38 (1.9)	156 (2.8)	194 (2.5)
Chronic diseases			
No	977 (48.3)	3098 (54.7)	4075 (53.0)
Yes	1046 (51.7)	2569 (45.3)	3615 (47.01)
Social activities			
No	732 (36.2)	3093 (54.6)	3825 (49.7)
Yes	1291 (63.8)	2574 (45.4)	3865 (50.3)
Having income or not			
No	1583 (78.3)	4493 (79.3)	6076 (79.0)
Yes	440 (21.7)	1174 (20.7)	1614 (21.0)

**Table 3 ijerph-18-03886-t003:** Determinants of depression in the elderly by binary logistics regression (n = 7690).

Variables	Unadjusted Model OR_-unadjusted_ (95% CI)	Fully Adjusted Model OR_-adjusted_ (95% CI)
Urban and rural source		
Urban	1	1
Rural	1.88 (1.67, 2.12)	1.52 (1.32, 1.76)
Gender		
Male	1	1
Female	1.95 (1.76, 2.15)	1.62 (1.43, 1.84)
Age		
60–64	1	1
65–74	1.04 (0.94, 1.16)	0.98 (0.86, 1.11)
≥75	1.02 (0.88, 1.19)	0.94 (0.79, 1.13)
Marital status		
Married and live with spouse	1	1
Married but not live with spouse	1.04 (0.80, 1.37)	0.98 (0.72, 1.33)
Divorced	1.29 (0.79, 2.09)	1.24 (0.69, 2.22)
Widower	1.60 (1.41, 1.81)	1.22 (1.05, 1.42)
Never married	1.32 (0.72, 2.42)	1.33 (0.66, 2.68)
Education level		
Illiterate	1	1
Elementary school and below	0.73 (0.65, 0.81)	0.93 (0.81, 1.07)
Middle school	0.43 (0.36, 0.50)	0.68 (0.56, 0.83)
High\vocational school and above	0.27 (0.22, 0.34)	0.47 (0.37, 0.61)
Minorities		
Han	1	1
Ethnic minorities	1.35 (1.12, 1.62)	1.31 (1.06, 1.62)
Religious beliefs		
No	1	1
Yes	1.06 (0.91, 1.24)	1.12 (0.94, 1.35)
Self-reported health		
Very good	1	1
Good	0.27 (0.02, 2.98)	0.41 (0.03, 5.34)
Fair	0.40 (0.04, 4.44)	0.56 (0.04, 7.19)
Poor	0.66 (0.06, 7.23)	0.75 (0.06, 9.56)
Very poor	2.02 (0.18, 22.28)	1.74 (0.14, 22.35)
Duration of sleep		
≤5 h	1	1
6–9 h	0.35 (0.31, 0.38)	0.48 (0.43, 0.54)
≥10 h	0.46 (0.38, 0.55)	0.53 (0.44, 0.65)
Life satisfaction		
Completely satisfied	1	1
Very satisfied	1.06 (0.80, 1.40)	1.09 (0.81, 1.46)
Somewhat satisfied	1.98 (1.51, 2.58)	1.98 (1.48, 2.64)
Not very satisfied	9.98 (7.27, 13.68)	7.39 (5.25, 10.39)
Not satisfied at all	29.80 (18.27, 48.78)	17.11 (10.11, 28.97)
Chronic disease		
No	1	1
Yes	1.63 (1.48, 1.79)	1.22 (1.09, 1.37)
Social activities		
No	1	1
Yes	0.73 (0.66, 0.80)	0.87 (0.77, 0.97)
Having income or not		
No	1	1
Yes	0.77 (0.69, 0.87)	0.80 (0.70, 0.92)

Note: (1) Taking the first category as the reference category (the one with minimum value). (2) OR, odds ratio, 95% odds ratio confidence interval.

**Table 4 ijerph-18-03886-t004:** Prevalence of depressive symptoms according to sociodemographic and health variables, urban and rural areas.

Variables	Urban (N = 2023, N (%))			Rural (N = 5667, N (%))		
	Depression	Normal	χ^2^-Value	Depression	Normal	χ^2^-Value
Gender						
Male	171 (17.3)	816 (83.7)	19.93 **	773 (26.1)	2189 (73.9)	173.60 **
Female	264 (25.5)	772 (74.5)		1155 (42.7)	1550 (57.3)	
Age						
60–64	155 (21.8)	557 (78.2)	3.70	692 (33.1)	1399 (66.9)	1.89
65–74	192 (20.0)	766 (80.0)		978 (34.9)	1825 (65.1)	
≥75	88 (24.9)	265 (75.1)		258 (33.4)	515 (66.6)	
Marital status						
Married and live with spouse	315 (20.0)	1259 (80.0)	13.67 *	1408 (32.0)	2997 (68.0)	49.78 **
Married but not live with spouse	17 (25.0)	51 (75.0)		63 (31.3)	138 (68.7)	
Divorced	5 (14.3)	30 (85.7)		20 (52.6)	18 (47.4)	
Widower	97 (28.5)	243 (71.5)		422 (42.9)	561 (57.1)	
Never married	1 (16.7)	5 (83.3)		15 (37.5)	25 (62.5)	
Education level						
Illiterate	69 (30.7)	156 (69.3)	35.45 **	720 (40.9)	1041 (59.1)	109.96 **
Elementary school and below	194 (25.5)	566 (74.5)		967 (34.1)	1882 (65.9)	
Middle school	10 0 (18.3)	447 (81.7)		186 (24.5)	572 (75.5)	
High \vocational school and above	72 (14.66)	41 9(85.34)		46 (15.9)	244 (84.1)	
Minorities						
Han	403 (21.4)	1482 (78.6)	0.25	1769 (33.5)	3519 (66.5)	11.38 **
Ethnic minorities	32 (23.3)	106 (76.8)		159 (42.0)	220 (58.0)	
Religious beliefs	393 (21.7)	1422 (78.3)				
No			0.24	1708 (33.8)	3343 (66.2)	0.88
Yes	42 (20.2)	166 (79.8)		220 (35.7)	396 (64.3)	
Self-reported health						
Very good	11 (5.2)	201 (94.8)	230.04 **	96 (11.9)	711 (88.1)	585.92 **
Good	25 (9.0)	253 (91.0)		144 (16.7)	720 (83.3)	
Fair	192 (17.9)	880 (82.1)		933 (24.7)	2849 (75.3)	
Poor	144 (41.0)	207 (59.0)		863 (50.2)	856 (49.8)	
Very poor	63 (57.3)	4 7(42.7)		327 (63.1)	191 (36.9)	
Duration of sleep						
≤5 h	240(35.5)	436(64.5)	121.78**	991(48.1)	1068(51.9)	288.07***
6–9 h	174(13.9)	1080(86.1)		755(25.5)	2200(74.5)	
≥10 h	21(22.6)	72(77.4)		182(27.9)	471(72.1)	
Life satisfaction						
Completely satisfied	17 (15.9)	90 (84.1)	246.16 **	54 (19.5)	223 (80.5)	605.07 **
Very satisfied	62 (10.2)	543 (89.8)		418 (22.3)	1455 (77.7)	
Somewhat satisfied	261 (22.2)	913 (77.8)		1004 (34.4)	1911 (65.6)	
Not very satisfied	61 (61.6)	38 (38.4)		317 (71.1)	129 (28.9)	
Not satisfied at all	34 (89.5)	4 (10.5)		135 (86.5)	21 (13.5)	
Chronic disease						
No	153 (15.7)	824 (84.3)	38.21 **	901 (29.1%)	2197 (70.9)	74.24 **
Yes	282 (27.0)	763 (73.0)		1027 (40.0%)	1542 (60.0)	
Social activities						
No	211 (28.8)	521 (71.2)	36.44 **	1093 (35.3)	2000 (64.7)	5.26 *
Yes	224 (17.4)	1067 (82.6)		835 (32.4)	1739 (67.6)	
Having income or not						
No	353 (22.3)	1229 (77.7)	3.03	1582 (35.2)	2911 (64.8)	13.66 **
Yes	82 (18.6)	359 (81.4)		346 (29.5)	828 (70.5)	

Note: (1) Chi-square tests (χ^2^-test) were used to compare the prevalence of depressive symptoms according to sociodemographic and health variables, urban and rural areas. (2) *: *p* < 0.05, **: *p* < 0.001.

**Table 5 ijerph-18-03886-t005:** Determinants and their effects on depression of the elderly in urban and rural areas by logistics regression.

Variables	Urban		Rural		Significance of Difference (Coefficient)
	OR	95% CI	OR	95% CI	
Gender					
Male	1	-	1	-	
Female	1.46 *	1.09, 1.96	1.68 **	1.46, 1.93	0.128
Age					
60–64	1	-	1	-	
65–74	0.81	0.59, 1.10	1.01	0.88, 1.16	0.191
≥75	0.99	0.66, 1.49	0.90	0.73, 1.11	−0.139
Marital status					
Married and live with spouse	1	-	1	-	
Married but not live with spouse	1.22	0.57, 2.59	0.94	0.66, 1.33	−0.150
Divorced	0.58	0.18, 1.85	2.15 *	1.01, 4.58	1.567 *
Widower	1.18	0.82, 1.72	1.24*	1.05, 1.47	0.106
Never married	0.48	0.04, 6.62	1.48	0.71, 3.07	1.086
Education level					
Illiterate	1	-	1	-	
Elementary school and below	0.82	0.54, 1.24	0.95	0.82, 1.10	0.097
Middle school	0.55 *	0.34, 0.88	0.71 *	0.56, 0.89	0.095
High\vocational school and above	0.40 **	0.24, 0.65	0.42 **	0.29, 0.61	−0.151
Minorities					
Han	1	-	1	-	
Ethnic minorities	1.11	0.67, 1.85	1.37 *	1.08, 1.74	0.200
Religious beliefs					
No	1	0	1	-	
Yes	1.10	0.90, 1.35	1.12	0.94, 1.34	−0.069
Self-reported health					
Very good	1	-	1	-	−0.179
Good	1.74	0.76, 4.00	1.35	0.97, 1.86	−0.298
Fair	2.50 *	1.22, 5.14	1.68 **	1.29, 2.19	−0.637
Poor	5.98 **	2.84, 12.59	3.92 **	2.98, 5.18	−0.625
Very poor	12.74 **	5.51, 29.46	5.71 **	4.08, 7.99	−0.849
Duration of sleep					
≤5 h	1	-	1	-	
6–9 h	0.42	0.33, 0.54	0.50 **	0.44, 0.58	0.181
≥10 h	0.66	0.37, 1.18	0.52 **	0.42, 0.65	−0.239
Life satisfaction					
Completely satisfied	1	-	1	-	
Very satisfied	0.69	0.34, 1.42	1.24	0.89, 1.74	0.707
Somewhat satisfied	1.55	0.79, 3.03	2.14 **	1.54, 2.98	0.357
Not very satisfied	6.57 **	2.90, 14.86	7.68 **	5.23, 11.29	0.118
Not satisfied at all	54.74 **	10.36, 289.30	15.34 **	8.61, 27.33	−0.650
Chronic disease					
No	1	-	1	-	
Yes	1.68 **	1.27, 2.23	1.17 *	1.03, 1.33	−0.243
Social activities					
No	1	-	1	-	
Yes	0.68 *	0.51, 0.90	0.93	0.82, 1.06	0.340 *
Having income or not					
No	1	-	1	-	
Yes	0.84	0.61,1.16	0.79 *	0.68, 0.93	−0.036

Note: (1) Taking the first category as the reference category (the one with the minimum value). (2) OR, odds ratio, 95% CI: 95% confidence interval. (3) Significance of difference (coefficient): interactions analyses between urban and rural sources and all variables are calculated to verify the interaction and check the impacts of the influencing factors. (4) *: *p* < 0.05, **: *p* < 0.001.

## Data Availability

All the data we used have been publicly released on the CHARLS website: http://charls.pku.edu.cn/pages/data/2018-charls-wave4/zh-cn.html (accessed on 28 December 2020).

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
