# Peer review of "Comparison of Depressive Symptoms and Its Influencing Factors among the Elderly in Urban and Rural Areas: Evidence from the China Health and Retirement Longitudinal Study (CHARLS)"

_ijerph, 2021, doi:10.3390/ijerph18083886_

Round 1

Reviewer 1 Report

This is, in summary, an interesting study aimed to compare depressive symptoms and its influencing factors among the elderly in urban and rural areas in a total of 7690 participants at age 60 or older. The authors found that there was a significant difference in the prevalence estimate of depression between urban and rural elderly. They also reported that the prevalence of depression among rural elderly was significantly higher than that of urban elderly. Importantly, after adjusting for gender, age, education level, marital status, self-reported health status, life satisfaction, chronic disease, sleep duration, social activities and income, the generalized linear model demonstrated the prevalence of depression in the rural elderly is 1.52 times that of urban elderly. Moreover, gender, education level, life satisfaction, chronic disease and social activities were influencing factors of depression among the urban elderly. Finally, for the rural elderly, influencing factors of depression resulted gender, education level, minorities, self-reported health status, life satisfaction, sleep duration (≤5h), chronic disease, social activities and income.

The authors may find as follows my minor comments/suggestions.

First, as throughout the Introduction section, the authors correctly reported that depression amongst the elderly is a global public health problem, they could even stress the link between depression and and sensory processing patterns. Here, the involvement of sensory perception which is implicated in the emotional processes of patients with depression, might be briefly discussed. Importantly, the unique sensory processing patterns of depressed individuals have been reported as crucial factors in determining unfavorable outcomes. Thus, given the above information, my suggestion is to include within the manuscript, the study published in 2016 on J Affect Disord (PMID: 28064114).

Moreover, as the main aims/objectives of the present study have been reported by the authors, the most relevant study hypotheses should be reported in a more detailed manner within the main text.

Furthermore, inclusion/exclusion criteria need to be extensively reported within the main text.

Also, the most relevant psychometric measures might be provided more succinctly.

Finally, what is the take-home message? While the authors reported that the prevalence of depressive symptoms in rural areas was higher than in urban areas, they failed to report the most relevant conclusive remarks of their paper. Specifically, which type of targeted intervention strategies are needed to prevent depression in urban and rural elderly? Here, more details/information are required for the general readership.

Reviewer 2 Report

Referee report on “Comparison of depressive symptoms and its influencing factors among the elderly in urban and rural areas: evidence from the China Health and Retirement Longitudinal Study”

Overall impression

The topic is interesting and important. This could be an interesting investigation, and especially the urban-rural perspective is important. Data used in this study is of high quality. However, there are some obvious weaknesses in the article. First, discussion on earlier studies and references to earlier studies concerning factors that influence depression is completely missing. Hence, the justification of the use of control variables is missing. There are also many inconsistencies in the article, and therefore it is unclear what is actually inspected in different parts of the paper. There are also inconsistencies in reporting the results (what factors are controlled for and what are not, there are also many mistakes in listing what variables are significant). The authors must make sure that they present the results correctly and coherently throughout the article. In general, results should be reported in more detail in the text. Presenting just a table or merely listing what factors are significant is not enough. Also, different parts of the paper use different reference categories. Finally, justification for the use of GLM in one investigation, but not in the other, is missing, and I believe it is unnecessary to use GLM. My more specific comments are listed below.

More detailed comments (not in any particular order)

In the abstract you write “Gender, education level, life satisfaction, chronic disease and social activities were influencing factors of depression among the urban elderly.” Self-reported health is also significant in urban areas.

In the abstract you write “But for the rural elderly, influencing factors of depression were gender, education level, minorities, self-reported health status, life satisfaction, sleep duration (≤5h), chronic disease, social activities and income (except for pension).” Social activities is not significant is rural areas.

In Figure 1, it is said that Table 4 is multivariate analysis. I believe this is not the case, Table 5 contains multivariate analysis. Does Table 3 really contain univariate analysis? See my comment later in this report.

Figure 1 is not complete, the results presented in Table 3 do not fit to any of the boxes.

Why have you selected those control variables that you use? Their selection should be justified, now there is no information why these particular variables are used.

Relating to this, references of earlier findings concerning factors that influence depression is completely missing. Findings of earlier studies should be summarized. This could be done when justifying the use of the control variables that you have selected or in the introduction.

All variables should be mentioned in the text. Chronic disease, at least few examples should be given. Having social activities, what does it include? What does Having social activities actually measure?

In Table 1, I suggest you put “Urban and rural areas” as the first variable in the table, since it is the main variable of interest.

2.3

First you write:

“Then a generalized linear model (GLM) was employed to estimate the association between living in urban or rural areas and depression after controlling other confounding factors (All variables in Table 1).”

Then you write:

“In our study, a binary logistic regression model was used to analyze the influence of living in urban or rural areas on depression in the elderly when controlling for other confounding factors.”

For me, this is contradictory. These sentences mean the same thing.  

It is also very unclear why you use generalized linear model in the first inspection, and logit model in second? This should be justified clearly. In particular, what is the value added using generalized linear model? In both investigations you have exactly the same dependent variable and exactly the same independent variables, so I don’t see any reason why you could not use logit model in both investigations. I think you should use logit model in all analyses.

Line 154-155 As far as I understand there is inconsistency in the definition of link function. In formula 3, you have g(µ) and in the following line you have g(π).

3.1

There is no text describing the characteristics of the participants (table 2). Presenting just a table is not enough, main characteristics of urban and rural participants should be summarized in the text.

In Table 2, the variables should be presented in the same way as in Table 3. For example, First row “Gender”, second row “Male”, third row “Female”.

3.2

line 174- It is unclear how you have obtained the unadjusted odds-ratio and the confidence intervals. Moreover, what is the added value of Figure 2? I don’t think Figure 2 is necessary. The unadjusted odds-ratio should be presented in Table 3 before presenting the fully adjusted model results (if the results presented are fully adjusted results, see my comment below).

You should show the prevalence of depressive symptoms in urban and rural areas in Table 4, so that the first line would be “Total”. In this case Figure 2 would not be needed.

If possible, Table 2 and Table 4 could be combined.

It is unclear, whether it is univariate analysis (as mentioned in your Figure 1) or multivariate analysis that is presented in Table 3. This needs clarification, and I think you should present multivariate results in Table 3.

Note in Table 3 “Note:OR, adjusted odds ratio and the adjusting variables (gender, age, education level, marital status, self-reported health status, life satisfaction, having chronic disease)” Why do you only list these variables as adjusting variables, even though in the text (line 180-181) you say you also control for “having social activities” and “having income” in the model? Yet a different set of control variables is mentioned in the abstract (line 23-24), where you write “After adjusting for gender, age, education level, marital status, self-reported health status, life satisfaction, chronic disease, sleep duration, social activities and income..”

You list three different sets of control variables, and therefore it is very unclear what factors you actually control for in the adjusted model.

I think all variables should be controlled for in the adjusted model, so control variables should also include “minorities”, “religious beliefs”, “duration of sleep”.

In Table 3, all significant values should be bolded. I think it is unnecessary to show P-values, you could use asterisks like you use in Table 5.

Relating to this, I think Table 3 and Table 5 should be combined. There is enough space to present all three models in the same table.

Line 182- “Gender, ethnic minorities, self-reported health status, life satisfaction, duration of sleep, chronic diseases, social activities and income (except pension) were also influencing factors of depressive symptoms in the elderly.” It seems to me that also education is significant.

Moreover, you have to explain in more detail in the text how these different factors are related to depressive symptoms. For example, women have higher likelihood of depressive symptoms.

You should also tell in the text what factors are not significant, because this information also might be important.

It seems quite strange to me that those who have no chronic diseases have higher risk of depression.

In all analyses, the group “never married” should be combined (preferably) with “divorced” since there are very few observations in these categories in the urban “depression” group.

3.3

Line 193- “The results of χ2 -test showed that in both urban and rural areas, the prevalence of depression varies significantly by gender, marital status, minorities, education level, self-reported health, life satisfaction, duration of sleep, chronic disease in the elderly (P < 0.05).” Minorities is not significant in urban areas p>0.05.

Line 197- “…for the elderly in urban areas, the prevalence of depression varies significantly by having social activities or not.” Having social activities is also significant in rural areas p<0.05

Line 198- “But for the elderly in urban areas, “having income or not (except pension)” was an important influencing factor of depression.” In urban areas p>0.05, but in rural areas p<0.001

In the models in Table 5, the educational groups should be combined in the same way in both urban and rural areas.

Line 200- and 207- “The results of binary logistic regression showed that the influencing factors of elderly depression in urban area were slightly different from those in rural area. For urban areas, gender…..” and “The impacts of those factors on depression were also different in urban and rural areas (Table 5).” First, you need to explain in the text how different variables are associated with depression in urban and rural areas. You also need to explain in the text what are the differences in the impacts.

Table 5. It seems to me that all of the classes of Self-reported health are significant in urban areas (CIs of ORs of Very good, Good, Fair are all under 1), yet you have not bolded them.

It also seems that you have combined age groups in the rural areas, since there is only one OR in addition to reference category. I don’t see any reason why you should combine age groups in rural areas.

Why have you changed the reference categories in Table 5, they are different from those in Table 3? Reference categories must be the same in all analyses. Reference categories in Table 5 seem to be in line with coding presented in Table 1 (for example in Table 1, religious beliefs, having income, chronic disease and having social activities are coded yes=1, no=0; the coding is similar in Table 5). Hence, Table 3 has to be changed.

Discussion

Line 237- “In both urban and rural areas, the prevalence of depression varies significantly among the elderly by characteristics, such as gender, marital status, chronic disease, education level, duration of sleep, self-reported health and life satisfaction” It is unclear if you refer to the results of other studies or if you refer to the results of your own study. If you refer to your own results, marital status is not significant in the models, and duration of sleep is only significant in rural areas. You need to clarify this part.

Line 258- “But in urban areas, the elderly who self-reported as “poor” health were 1.49 times more likely to suffer from depression than those who self-reported as “very poor” health, and the effects of other self-reported health levels on depression were similar to the “very poor” level in the elderly.” Based on the results in Table 5 this is incorrect interpretation.

In general, I miss more discussion on the possible reasons why different factors are differently related to depression in rural and urban areas.

Round 2

Reviewer 1 Report

In the revised paper, the authors addressed most of the major questions raised by Reviewers improving both the main structure and quality of this paper. I have no further comments.

Author Response

Thank you very much for your valuable suggetions in the first round of revision!

Reviewer 2 Report

Referee report 2

The paper has improved somewhat. However, even if the authors state that they have corrected the mistakes that I pointed out in my previous comments, and assure they won’t do such a “low-level mistake” again, some of the mistakes have not been corrected. This makes me wonder if the authors really want to get their work published. If in future versions I see same kinds of mistakes, I will not be happy. So I suggest that the authors very carefully check their manuscript (both tables and text) before submitting it again. Moreover, the authors now present hypotheses, and in order to answer there, additional analyses are required.

Abstract:

In the abstract you write “(urban: 21 435/2023=21.5%; rural: 1928/5667=34.0%).” Why do you present these numbers and why have you excluded the unadjusted odds-ratio from the abstract? I think you should present the unadjusted odds-ratio, in addition to the shares.

It would be easier to see the differences between urban and rural areas, if you first list those factors that are common to both urban and rural areas, and after that would add what other factors are significant in urban and rural areas. For example, “Gender, education level, self-reported health, life satisfaction, chronic disease were associated with depression in both urban and rural areas. In addition, social activities were connected with depression in urban areas, while minorities, duration of sleep and income were influencing factors of depression among the rural elderly.”

Introduction

This sentence is too vague “Some studies indicated that depressive symptom of elderly was also associated with gender, age, education level, sleep, chronic diseases, social activities and so on. ” You cannot just put “and so on” to the sentence describing the findings of earlier studies.

In your response you say “According to the literature study, gender, age, marital status, nationality, education level, self-rated health, sleep, life satisfaction, chronic diseases, social activities and income may be the influencing factors of depression in the elderly.” But in your literature review you don’t mention any studies that link education level, self-rated health, life satisfaction and income with depression. Examples of these should be given. Moreover, you need to be more specific on the direction of connection, how different factors have been found to be connected with depression. For example how is age, sleep duration and other factors connected with depression. Is older age connected with higher or lower risk of depression etc.

You write “Hypothesis 2. The influencing factors of depression symptoms are different between urban and rural elderly, and the impacts of these factors on depression symptoms are different too.” This needs to be clarified. In your first sentence you say that influencing factors are different in urban and rural areas. Then of course the impact of these factors are different. You need to be more specific in your second statement. For example: Even if the influencing factors are the same, the impacts of these factors on depression symptoms are different in urban and rural areas.

2.3

You write “Due to the score of depression was not the normal distribution, a generalized linear model (GLM) was employed to estimate the association between urban and rural sources and depression, also was employed to analyze the influencing factors of depression in urban and rural elderly respectively.” It doesn’t matter that the score of depression is not normally distributed, since you use binary dependent variable. So I very strongly suggest you simply use logistic regression in all analyses and describe the method accordingly. The results are anyway the same. So I think you must totally reject the idea and explanation for GLM (there is no need to present things so complicated since you actually use logistic regression) and just simply say that you use logistic regression in your statistical analysis. This is my very strong recommendation.

Table 2. It is good that you have added explanation on the contents of Table 2 to the text. However, presenting exact %-figures is against good writing practices. There is no need to repeat the exact %-figures in the text.

NOT ”Among the 7690 participants, urban elderly accounted for about 26.3%, and urban elderly accounted for 73.7%. In urban areas, male accounted for 48.8%, 47.4% elderly were 65 to 74 years old and 35.2% were 60 to 64 years old, Married and live with spouse accounted for 77.8%, 53.0% of the elderly’s self-reported health were fair and 22.8% elderly’s self-reported health were poor and very poor, 6.8% of the elderly were not very satisfied and not at all, and 51.7% of the elderly had chronic disease. While in rural areas, male accounted for 52.3%, 49.5% elderly were 65 to 74 years old and 36.9% were 60 to 64 years old, married and live with spouse accounted for 77.7%, 47.8% elderly’s self-reported health was fair and 31.3% elderly’s self-reported health was poor and very poor, 10.7% of the elderly were not very satisfied and not at all, and 45.3% of the elderly had chronic disease.”  

BUT rather: Among the 7690 participants, urban elderly accounted for about 26%, and urban elderly accounted for 74%. In urban areas, there were slightly less men than women, while the situation in rural areas was the opposite. In both areas, around half of the elderly were 65 to 74 years old and around one third were 60 to 64 years old. The share of oldest age group was somewhat larger in urban areas. In both areas married and live with spouse accounted about 75%. Around half of the of the elderly’s self-reported health were fair in both areas, and around one fifth of elderly’s self-reported health were poor and very poor in urban areas, in comparison to one almost one third in rural areas. In urban areas around 7% of the elderly were not very satisfied and not at all satisfied, while in rural areas this number was 10%. Slightly more than 50% of urban elderly had chronic disease, and the share of rural elderly with chronic disease was around 45%.

3.2

Why have you selected not to present the unadjusted odds-ratio? I think it should be presented. This is a common way to present results: first you only control for urban and rural origin in the model (unadjusted odds-ratio), second you add other control variables to the model (adjusted odds-ratio).  It gives more information than % shares and χ2 test. % shares and χ2 test can be presented but unadjusted odds-ratio should also be presented in table.

Variables

Unadjusted model

Fully adjusted model

OR

OR

Urban

1

1

Rural

X.XX

1.52

Other variables

X.XX

In your response to point 12 you write “Through literature research, we also found that the effects of divorce and never married on depression of the elderly were different. (Li T., Chen XZ., Yin SF., Zhang LC. Influence of marital status on life satisfaction and depression of empty nest elderly[J]. Chinese Journal of Gerontology,2018,38(16):4058-4059.Zhou J., Liu Y., Yuan H., et al. The Prevalence and influencing factors of depressive symptoms among the elderly in a rural community of Anhui province[J]. Journal of Qiqihar University of Medicine,2018,39(5):573-576. Pei QY. Analysis of depressive symptoms and influencing factors in Chinese elderly[J]. Zhengzhou University,2019,06.).” Why do you not mention these results in your literature review in the introduction? This is an important detail.

My earlier point 16 still holds. “Point 16Why have you changed the reference categories in Table 5, they are different from those in Table 3? Reference categories must be the same in all analyses. Reference categories in Table 5 seem to be in line with coding presented in Table 1 (for example in Table 1, religious beliefs, having income, chronic disease and having social activities are coded yes=1, no=0; the coding is similar in Table 5). Hence, Table 3 has to be changed.” In your response you say “Response 16According to your suggestion, we have revised the reference categories in Table 1, Table 2, Table 3, Table 4 and Table 5 to make their expressions consistent, also the expressions throughout the text.”

You have NOT revised the reference categories and the expression are NOT consistent. In Table 3, religious beliefs, having income, and having social activities are coded yes=0, no=1, in Table 5 (and in Table 1) these variables are coded no=0, yes=1.

My earlier point 11 still holds. “Point 11It seems quite strange to me that those who have no chronic diseases have higher risk of depression.” In your response you say "Response 11We are so sorry for this low-level mistake that we made when we filled out Table3.  We will not make such low-level mistake again. In fact, the prevalence of depression in the elderly with chronic diseases was higher than that in the elderly without chronic diseases. This result is consistent with that in Table 4. The prevalence of depression in the elderly with chronic diseases is 1.22 times that in the elderly without chronic diseases."

You have not changed this. The results relating to “Chronic disease” in Table 3 and Table 5 are still presented incorrectly. In both tables you have yes=0, no=1, so the results in both tables show that NOT having chronic disease is associated with higher odds of depression. I don’t understand how it is so hard to get these things right. The authors must make sure that the results are presented correctly.

Also the result relating to social activities seems to be incorrect. Now your results in Table 3 tell that those WITHOUT social activities have lower risk of depression.

The order of  “Male” and “Female” is incorrect in Table 5.

In my earlier point 16 I only gave you examples what variables are inconsistent in Tables 3 and 5. In addition, there were other variables that were inconsistent in these tables, but I didn’t think that I must mention separately EVERY variable that is coded inconsistently in Table 3 vs. Table 5. The authors must make sure that reference categories are the same and the order of variable classes is the same in all tables. Now they are not. This shows that the authors have not carefully checked their results. It is not my job as a referee to check these kind of sloppy mistakes and inconsistencies!!!!

Variable

Table 3

Table 5

Age

Reference category: 60-64

Reference category: >=75

Marital status

Reference: Married and live with spouse

Reference: Never married

Education

Reference: Illiterate

Reference: High/vocational school and above

Minorities

Reference: Han

Reference: Ethnic minorities

Religious beliefs

Reference: Yes

Reference: No

Self-reported health

Reference: Very good

Reference: Very poor

Duration of sleep

Reference: <=5h

Reference: >=10h

Life satisfaction

Reference: Completely satisfied

Reference: Not satisfied at all

Social activities

Reference: Yes

Reference: No

Having income

Reference: Yes

Reference: No

Moreover, in your reply you say that you tried combining Tables 3 and 5. I don’t believe you really tried it, since you did not notice above mentioned differences in the reference categories. But I accept that they are kept separate.

You write “Women had higher likelihood of depressive symptoms (OR=1.52, 95% CI:1.32 to 1.76), the prevalence estimates of Ethnic minorities was higher than Han minorities (OR=1.31, 95% CI:1.06 to 241 1.62). The poorer the self-rated health, the higher the prevalence of depression in the elderly, the lower the lift satisfaction, the higher the prevalence of depression in the elderly. The prevalence of depression in the elderly with chronic diseases was higher than that in the elderly without chronic diseases (OR=1.22, 95% CI:1.09 to 1.37), the prevalence of depression in the elderly without social activities than that in the elderly having social activities (OR=0.87, 95% CI:0.78 to 0.97), the prevalence of depression in the of depression in the elderly, the lower the lift satisfaction, the higher the prevalence of depression in the elderly. The prevalence of depression in the elderly with chronic diseases was higher than that in the elderly without chronic diseases (OR=1.22, 95% CI:1.09 to 250 1.37), the prevalence of depression in the elderly without social activities than that in the elderly having social activities (OR=0.87, 95% CI:0.78 to 0.97), the prevalence of depression in the elderly without income than that in the elderly having income (OR=1.25, 95% 253 CI:1.08 to 1.43).” Something wrong in this text?

The effect of Education level, duration of sleep should also be mentioned.

3.3

Note in Table 4 “Therefore, in the logistic regression analyses, we merged groups “Bachelor degree and above” and “High\vocational school and associate degree” of the elderly.” You are not presenting logistic regression analysis in this Table, so the word “logistic” needs to be deleted.

You write “The results of GLM showed that the influencing factors of elderly depression in urban area were different from those in rural area. For urban areas, gender, education level (illiterate), self-reported health, life satisfaction, chronic disease and social activities were the influencing factors of elderly depression. But for rural areas, the influencing factors of depression included gender, education level, minorities, self-reported health, life satisfaction, duration sleep (≤5h), life satisfaction, chronic diseases and having income or not. And the impacts of those factors on elderly depression were different in urban and rural areas”

As I write in my comment regarding the abstract, I think it would be easier to see the differences between urban and rural areas, if you first list those factors that are common to both urban and rural areas, and after that would add what other factors are significant in urban and rural areas. For example, “Gender, education level, self-reported health, life satisfaction, chronic disease were associated with depression in both urban and rural areas. However, the impacts of these factors were somewhat different. In urban areas, the prevalence of depression in female elderly was 1.43 times (OR=1.43, 95% CI: 1.13 to 1.93) than that of male elderly, while in rural areas, the prevalence of depression in female elderly was 1.67 times (OR=1.67, 95% CI: 1.45 to 1.92) than that of male elderly…….” I think it should also be mentioned that in general, those with better self-reported health have lower odds of depression on both areas but there is no difference between those with very poor and very good health in rural areas. Higher life satisfaction is associated with lower risk of depression in both areas, while having chronic disease is connected with increased odds of depression.

After that you would describe what other factors are significant in urban and rural areas, for example:

“In addition, social activities were connected with depression in urban areas, while minorities, duration of sleep and income were influencing factors of depression among the rural elderly. Urban residents with social activities had lower odds of depression. In rural areas, those belonging to Han had higher risk of depression (I believe this result is presented incorrectly in Table 5, I think it should be those belonging to Ethnic minorities have a higher risk of depression). Likewise, those with shorter duration of sleep had a higher likelihood of depression, while those having income had a lower risk of depression. ”

Relating to your statement that “And the impacts of those factors on elderly depression were different in urban and rural areas”. Have you checked that the differences you mention in the text (gender, education) are significant? Are you sure that the effect of other variables are not significantly different in urban and rural areas? For example it seems that the effect of life satisfaction is stronger in urban areas, and so is the effect of having chronic disease. You need to check the significance of all differences, since in your hypothesis 2 you say that the impacts of influencing factors are different between urban and rural areas. Otherwise you cannot answer to the hypothesis. Checking if the differences are significant can be done by using interactions between rural origin and all variables in the model presented in Table 3. In order to answer your hypothesis 2 you need to check these. You can report the results relating to interaction variables (difference significant or not) in the additional column in Table 5.

Variable

Urban

Rural

Significance of difference

Gender

Male

Female

X.XX

X.XX

X.XX

X.XX

**

Relating to your hypotheses, you do not clearly state the answer to your hypotheses in the text. Answers need to be stated (that is, was the hypothesis supported or not).

Discussion

You write “But in urban areas, the prevalence of depression among illiterate was 2.87 times that of the elderly with a bachelor degree or above.” This in incorrect, based on Table 5. Moreover, from Table 5 it appears that education has stronger impact in rural areas.

You  write “In our study, we found that after controlling other independent variables, in urban and rural areas, the poorer the self-reported health, the higher the prevalence of depression in the elderly, not only in urban areas, but also in rural areas.” From Table 5 it appears that in rural areas there is no difference between “very poor” and “very good”. This has also be mentioned and explanations for this must 
